# The association of COVID-19 lockdowns with adverse birth and pregnancy outcomes in 28 high-income countries: a systematic review and meta-analysis

We conducted a systematic review and meta-analysis to review the association of lockdowns with adverse birth and pregnancy outcomes (ABPOs) and related inequalities, in high-income countries (HICs). Databases (EMBASE, MEDLINE/PubMed and Web of Science) were searched from 1 January 2019 to 22 June 2023 for original observational studies based in HICs that compared the rates of ABPOs, before and during lockdowns. The risk of bias was assessed using the Newcastle–Ottawa tool for cohort studies. We ran random-effects meta-analyses and subgroup analyses per region, lockdown period, ethnicity group and deprivation level and adjusted for underlying temporal trends. A total of 132 studies were meta-analysed from 28 HICs. Reduced rates of preterm birth (reported by 26 studies) were associated with the first lockdown (relative risk 0.96, 95% confidence interval 0.93–0.99), 11 studies adjusted for long-term trends and the association remained (0.97, 0.95–0.99), and subgroup analysis found that this association varied by continental region. Ten studies reported positive screening rates for possible depression antenatally, and lockdown was associated with increases in positive screening rates (1.37, 1.06–1.78). No other ABPOs were associated with lockdowns. Investigation of inequalities was limited due to data availability and heterogeneity; further research is warranted on the effect of lockdowns on health inequalities. This study was funded by the National Institute of Health Research, School of Primary Care Research and registered on PROSPERO (CRD42022327448).

In early 2020, the global spread of severe acute respiratory syndrome coronavirus 2 (leading to coronavirus disease 2019 (COVID-19)) prompted governments around the world to announce a variety of non-pharmaceutical interventions, largely referred to as lockdowns, to reduce infection rates[1,2]. There is increasing evidence that lockdowns were associated with changes in adverse birth and pregnancy outcomes (ABPOs), which were not directly related with the COVID-19 infection and are usually considered an indirect effect of lockdowns[3–6].

Initial studies reported heterogeneous associations with ABPOs and lockdowns that varied according to country income level; these included decreases in preterm birth (PTB) in high-income countries (HICs) and increases in stillbirth in low-to-middle income countries (LMICs)[3–7].

The associations between ABPOs and lockdowns in HICs needs to be examined separately from LMICs, as each setting differed in lockdown measures, healthcare systems, COVID-19 infection and

✉e-mail: i.hindes@qmul.ac.uk

testing rates. Even within HICs, there was considerable variability in the implementation of lockdown measures across countries, which could have led to different associations with ABPOs[1,8]. Furthermore, the nature of these lockdown measures evolved over time, with distinct phases of the pandemic marked by the first and second lockdowns[8]. Therefore, associations with ABPOs were not consistent during and between lockdowns[7,9,10]. To gain a better picture of how lockdowns were associated with ABPOs, analysis by region and defined period of lockdown is necessary.

ABPOs are unequally distributed between ethnic groups and deprivation levels within HICs, heavily burdening minorities and those living in high deprivation[11–13]. Lockdowns probably exacerbated such inequalities; however, the evidence from HICs is conflicting[8,14]. In the UK, national data suggest that lockdowns are associated with worsened inequalities in ABPOs between ethnicity groups, but inequalities in PTB rates between deprivation quintiles may have been unaffected or even associated with reductions[4,5]. The associations between lockdowns and inequalities in ABPOs within HICs are unclear, and research is required to develop guidance and policies that would enable health equity in the future[8,14].

Lockdowns are associated differently with ABPOs in HICs compared with LMICs. While inequalities in ABPOs are present in both LMICs and HICs, within HICs, there is accumulating evidence that associations with lockdowns were inconsistent between regions, time periods, ethnicity groups and deprivation levels. The aim of this systematic review and meta-analysis is to assess the evidence on the associations between COVID-19 lockdowns and ABPOs in HICs and whether any associations differed by ethnicity or deprivation, timing of lockdown or continental region.

## Results

The database search yielded 14,215 results. After deduplication, 9,870 titles and abstracts were screened. A total of 416 met the criteria for full text screening, among which 207 were identified as eligible studies. After accounting for studies that had a cohort overlap (different studies that overlap in population, outcome and time periods) (31 studies), and studies with missing, unreported, illegible or wrongly formatted data (44 studies), 132 studies were included in the meta-analysis (Supplementary Material 4)[5,7,9,10,15–143]. Eleven were included in time-adjusted analysis (Supplementary Material 5)[7,10,28,29,52,67,117,144–147]. The study selection process is summarized in a Preferred Reporting Items for Systematic Reviews and Meta-Analyses (PRISMA) flowchart (Supplementary Material 6). Most papers included in this review had low risk of bias, although two studies exploring maternal mental health scored poorly on the bias assessment (Supplementary Materials 4 and 5). Twenty-eight countries were included; the countries with the most studies were the USA (46 studies), Italy (13 studies) and Canada (11 studies). Europe and North America were overrepresented in the dataset (59 studies (45%) and 54 studies (41%), respectively). There were fewer studies from Asia (10 studies, 7.6%), the Middle East (11 studies, 8.3%), Oceania (5 studies, 3.8%) and South America (1 study, 0.8%). Most studies were cohort studies (114 studies, 86%), followed by cross-sectional studies (15 studies, 11%), two prevalence proportion studies and a single case–control study. Half of the studies were based on data from single sites, such as individual hospitals or clinics (66 studies, 50%). Thirty-eight studies used regional- or state-level data (29%), and 28 used national datasets (21%).

PTB was reported in 26 studies with a total of 3,330,921 livebirths during lockdown and 31,969,791 livebirths in the pre-lockdown cohort. Overall, there was a 4% decrease in the risk of PTB associated with lockdown, compared with pre-lockdown (relative risk (RR) 0.96, 95% confidence interval (CI) 0.93–0.99) (relative risk is a ratio which compares the risk of an event (in this case an ABPO) in two groups (in this case a group exposed to lockdown restrictions and a group unexposed to lockdown restrictions); 95% confidence interval is the range of values that

is likely to contain the true population value) (Supplementary Fig. 1a). There was high heterogeneity between studies ($I^2 = 96.1\%$, $P < 0.001$), so regional subgroup differences were explored. Subgroup analysis found that the associations between PTB and lockdown differed according to continental region (Supplementary Fig. 1b, $P < 0.001$). Lockdown was associated with decreases in PTB in Europe (RR 0.94, 95% CI 0.92–0.96) and Australia (RR 0.94, 95% CI 0.90–0.99), but there was no credible evidence of an association between PTB and lockdown in North America (RR 1.01, 95% CI 1.00–1.01), the Middle East (RR 0.91, 95% CI 0.81–1.03), Asia (RR 1.02, 95% CI 0.89–1.18) or South America (RR 1.01, 95% CI 0.96–1.06) (Supplementary Fig. 1b). Fourteen studies on PTB reported data stratified according to ethnicity and deprivation. The test for subgroup differences suggests that there is no evidence of a subgroup effect by ethnicity ($P = 0.268$) or deprivation ($P = 0.503$) on PTB (Supplementary Fig. 1c,d). Nine studies included time-adjusted estimates of PTB in the first lockdown. Calvert and colleagues reported time-adjusted estimates for multiple populations; each estimate was included with the population in brackets. When underlying temporal trends of PTB were adjusted for, there was still a decrease of 3% associated with the lockdown period (RR 0.97, 95% CI 0.95–0.99) (Supplementary Fig. 1e).

Supplementary Material 7 includes subgroup analyses on PTB and shows that there was insufficient evidence to support an association between the risk of PTB and the second lockdown (RR 0.95, 95% CI 0.90–1.00) or post-lockdown (RR 0.98, 95% CI 0.93–1.04) compared with pre-lockdown. Supplementary analysis also found that spontaneous PTB was associated with a decrease during lockdown (RR 0.95, 95% CI 0.90–0.99), whereas evidence was insufficient to support an association between the risk of iatrogenic PTB and lockdowns (RR 0.95, 95% CI 0.88–1.03) (Supplementary Material 7).

Twenty-six studies, including 2,424,824 births in lockdown and 9,101,854 births pre-lockdown, reported stillbirth. Meta-analysis found no credible evidence of an association between stillbirth and lockdown (RR 1.04, 95% CI 1.00–1.08) (Supplementary Fig. 2a). Five studies reported time-adjusted estimates of stillbirth during lockdown. No credible evidence supported an association between stillbirths and the lockdown period when underlying temporal trends were considered (RR 1.02, 95% CI 0.98–1.06) (Supplementary Fig. 2b).

Low birth weight (LBW), reported by 18 studies, had insufficient evidence to support an association with lockdown (RR 0.97, 95% CI 0.93–1.01) (Supplementary Fig. 3a). Due to high heterogeneity ($I^2 = 81.9\%$, $P < 0.001$), regional subgroups were analysed, but the test for subgroup differences suggested that there was no evidence to support that the association differed by region ($P = 0.292$) (Supplementary Material 7). Neonatal mortality, reported by nine studies, was associated with an 18% decrease during lockdown (RR 0.82, 95% CI 0.74–0.91) (Supplementary Fig. 3b). Neonatal admissions, reported by 23 studies, had no credible evidence to support an association with lockdown (RR 1.03, 95% CI 0.98–1.08) (Supplementary Fig. 3c); however, due to considerable heterogeneity ($I^2 = 88.2\%$, $P < 0.001$) regional subgroup analysis was run. The test for subgroup differences supported that the association differed by region ($P = 0.018$), in which a decrease in neonatal admissions was associated with lockdown in the Middle East (RR 0.84, 95% CI 0.72–0.97), an increase in neonatal admissions was associated with lockdown in Asia (RR 1.38, 95% CI 1.05–1.83), and no credible evidence supported associations between lockdown and neonatal admissions in North America (RR 1.03, 95% CI 0.97–1.10), Europe (RR 1.03, 95% CI 0.96–1.10) or Australia (RR 1.00, 95% CI 0.90–1.10) (Supplementary Material 7). Hypoxic ischaemic encephalopathy was reported in only one study that found insufficient evidence to support an association with lockdown[94]. Prolonged neonatal stay in hospital was reported by two studies; both studies reported lower rates of prolonged neonatal stay in the lockdown[110,116].

Caesarean section was reported by 25 studies and did not have sufficient evidence to indicate that it was associated with lockdown (RR 1.01, 95% CI 1.00–1.03) (Supplementary Fig. 4a). Due to high

heterogeneity between studies in the overall caesarean section meta-analysis ($I^2$ = 82.8%, $P$ < 0.001), subgroup analysis by region was run. The test for subgroup differences suggested no evidence of a subgroup effect by region ($P$ = 0.189) (Supplementary Material 7). Evidence was also insufficient to support an association between lockdown and subclassifications of caesarean section (emergency (RR 0.99, 95% CI 0.97–1.02) or planned (RR 1.03, 95% CI 0.99–1.07)). Obstetric anal sphincter injuries (OASI) were reported by five studies, and peripartum hysterectomy was reported by four, neither of which had sufficient evidence to support an association with lockdown (RR 1.02, 95% CI 0.93–1.11 and RR 0.89, 95% CI 0.60–1.31, respectively) (Supplementary Fig. 4b,c).

Sepsis was reported by two studies, neither of which reported a change in risk or sufficient evidence to support an association with lockdown[127,135]. Prolonged maternal hospital stay was reported by three studies, but it was not meta-analysed as it was below the study limit for meta-analysis and outcome definitions varied considerably between studies; all studies reported decreased risk of prolonged maternal stay associated with lockdown[5,43,110].

Maternal readmission was reported by nine studies and had no credible evidence to support an association with lockdown (RR 0.98, 95% CI 0.79–1.21) (Supplementary Fig. 4d). Due to considerable heterogeneity ($I^2$ = 97%, $P$ < 0.001), regional subgroup analysis found associations between maternal readmission and lockdown differed by continental region ($P$ < 0.001). There was a decrease in maternal readmission in Europe only, specifically the UK, associated with lockdown compared with pre-lockdown (RR 0.69, 95% CI 0.66–0.71). No credible evidence supported an association with lockdown in North America (RR 1.02, 95% CI 0.99–1.05) or the Middle East (RR 1.40, 95% CI 0.67–2.90) (Supplementary Material 7). Meta-analysis of ten studies explored positive screening for depression antenatally with questionnaires. The risk of screening positive (clinically relevant scores) for depression antenatally was associated with a 37% increase during lockdown compared with the pre-pandemic period (RR 1.37, 95% CI 1.06–1.78) (Supplementary Fig. 4e). However, evidence was insufficient to support an association between lockdown and screening positively for depression postpartum (RR 1.01, 95% CI 0.90–1.14) (Supplementary Fig. 4f). There was no evidence to support an association between the risk of screening positive for maternal anxiety antenatally and lockdown (RR 1.08, 95% CI 0.81–1.44) (Supplementary Fig. 4g). Heterogeneity was considerable for both positive screening for depression and anxiety antenatally ($I^2$ = 75.9%, $P$ < 0.001, and 89.4%, $P$ < 0.001, respectively), so subgroup analysis by continent was run for each. There was insufficient evidence to support that associations differed by continental region for either outcome (anxiety antenatally $P$ = 0.797 and depression antenatally $P$ = 0.662) (Supplementary Material 7). Three studies reported on positive screening for anxiety postpartum, below the limit for meta-analysis, but all studies reported insufficient evidence to support an association with lockdown[46,78,140].

### Sensitivity analyses

When only the four studies that stratified data according to area deprivation composite indices were included in the analysis of PTB and lockdown according to deprivation level, the test for subgroup differences according to deprivation level again found no evidence of an effect by deprivation on the association between PTB and lockdowns ($P$ = 0.201) (Supplementary Fig. 5).

The possibility of publication bias was explored in outcomes that had over ten studies meta-analysed, including PTB, stillbirth, LBW, neonatal admissions, caesarean section and screening for depression postpartum. There was no evidence of publication bias or small study effects for any pooled estimates (Supplementary Fig. 6a–f).

The statistical power of samples to detect the pooled estimate obtained from meta-analysis and the minimum detectable effect of each meta-analysis was investigated in all outcomes and

subgroups that had insufficient evidence to support an association with lockdown.

The following outcomes and subgroups were sufficiently powered (>80%) to investigate the pooled effect estimate obtained by meta-analyses. Each outcome or subgroup is reported with the estimated power of the sample to investigate the pooled estimate and minimum detectable risk ratio, both in parentheses.

- PTB subgroup analyses by region:
  - North America (power to investigate subgroup pooled estimate: 98%, minimum detectable risk ratio of subgroup: 1.0069),
  - Middle East (94%, 0.9276).
- PTB subgroup analyses by deprivation:
  - Medium deprivation (83%, 0.9518)
  - High deprivation (81%, 0.9801).
- Stillbirth (97%, 1.0287)
- LBW (100%, 0.9926)
- Neonatal admissions (100%, 1.0148)
- Caesarean section (100%, 1.0033)

The following outcomes and subgroups were not sufficiently powered (<80%) to investigate the pooled effect estimate obtained by meta-analyses.

- PTB subgroup analyses by region:
  - Asia (60%, 1.0254)
  - South America (7%, 1.0671)
- PTB subgroup analyses by ethnicity:
  - Black (10%, 0.9588)
  - Other (52%, 0.9176)
  - Hispanic (12%, 0.9628)
  - Asian (76%, 0.9581)
- OASI (7.65%, 1.1235)
- Peripartum hysterectomy (17%, 0.7171)
- Maternal readmission (54%, 0.9728)
- Positive screening for depression postpartum (11%, 1.0394)
- Positive screening for anxiety antenatally (55%, 1.1079)

## Discussion

Our review provides an assessment of the association of lockdowns with ABPOs and inequalities in HICs. We provided evidence that PTB was associated with decreases during the first lockdown (by 4%) even after accounting for temporal trends (by 3%), which is consistent with other reviews. However, we provide evidence that the associated decrease was evident only in Europe and Australia[3,6,7]. Supplementary analyses showed that lockdowns were associated with decreases in spontaneous PTB and PTB, which occurred at moderate-to-late gestational age (Supplementary Material 7).

It is widely accepted that ABPOs, particularly PTB, are influenced by socio-environmental or modifiable lifestyle risk factors, including social support, work conditions and access to care[13,148–155]. Many such determinants were dramatically affected during lockdowns and may contribute to the association between lockdowns and decreases in PTB. Before the pandemic, these socio-environmental determinants of health were unequally distributed between ethnic groups and deprivation levels, resulting in substantial inequalities in ABPOs[11,156]. Lockdowns could have had a synergistic effect and compounded such inequalities[14]; ethnic minorities and those in the most deprived areas were hit harder by restrictions, suffered increased barriers to care, had higher morbidity and mortality rates from COVID-19 and had higher rates of job loss and financial insecurity[157–164], whereas individuals with higher financial and social security probably benefitted from the flexibility of remote work conditions[8]. However, this review

had insufficient evidence to support that the association between lockdown and decreased PTB varied according to ethnicity group or deprivation level.

The investigation of inequalities in ABPOs between ethnicity and deprivation groups was limited due to data availability and heterogeneity between studies and settings. Very few studies reported data stratified according to ethnicity and deprivation, and fewer used the same ethnicity groups or deprivation levels. This is understandable as ethnicity distribution varies between settings, and deprivation indicators such as maternal education and neighbourhood income are contextually specific. We addressed this limitation pragmatically by synthesizing available data on deprivation or by using proxy variables for deprivation, and harmonizing composite and proxy variables to obtain aggregated estimates of deprivation. Data synthesis meant that all subgroups in the analyses of deprivation had sufficient power to detect the pooled estimate obtained in meta-analyses. In the subgroup meta-analyses of PTB by ethnicity, only the white ethnicity group had sufficient power to detect the pooled estimate. All non-white ethnicity groups were insufficiently powered, and further research is required on larger, more consistent data to estimate these subgroups' association with lockdown.

In terms of deprivation, area-level estimates of deprivation have an inherent limitation of misclassification of individuals, for instance, those who live in areas of high deprivation but are not themselves deprived; this non-differential misclassification blunts the gradient of observed differences in deprivation. When these estimates are combined, this is compounded, underestimating the association which biases towards the null. Our results indicate that lockdown's association with PTB did not vary according to socio-economic deprivation and ethnicity; however, the magnitude of the association may be underestimated and analysis using individual deprivation indices is warranted.

Access to high-quality maternity care is a key determinant of maternal and neonatal health, which was considerably altered by lockdown restrictions[149,165]. Authors hypothesize about the causal mechanisms in the relationship between lockdowns and ABPOs, such as the influence of lockdowns on the delivery of and access to maternity services. However, due to data limitations and lack of sufficient evidence, the association between lockdowns and the supply and demand of maternity care could not be explored in greater detail in this review.

Nevertheless, the surrounding literature indicates that in HICs maternity services largely remained open throughout lockdowns, but services were altered to protect patients and practitioners from infection risks; care alterations in HICs included remote consultations via telephone and home blood pressure monitoring[158,165]. During the pandemic, there were decreases in antenatal visits and screening uptake, indicating that the lockdowns substantially influenced care-seeking behaviour among pregnant women[165]. It appears that the influence of these service adaptations varied according to an individual's income level and resource[165]. The multifaceted influences of the pandemic and lockdowns on maternity service provision and care-seeking behaviour requires further investigation to ascertain how these care-related factors may have affected ABPOs.

We identified a decrease of 18% in neonatal mortality associated with the lockdown[3,6]. However, the neonatal mortality estimate was dominated mainly by a study by Shukla and colleagues, weighted 60%; in their study, when previous long-term trends in neonatal mortality were accounted for, they found no substantial change during lockdown[121]. Therefore, the estimated 18% decrease in the risk of neonatal mortality should be interpreted with caution and emphasizes the need for further investigation using time-adjusted analysis to precisely assess the association with lockdown measures[3,6].

Our review aggregates data on positive screening scores for maternal mental health outcomes during lockdown. While previous reviews have reported increased perinatal depression associated with the pandemic, our pooled estimates found evidence to support this association antenatally, but no credible evidence of an association between positive screening rates of postpartum depression and lockdowns[166–168]. Other studies have suggested that limited access to care, multidisciplinary support and treatment in lockdowns aggravated depression among both pregnant and postpartum women[166,167]. It is important to note there was variation in measurement modalities and timing of screening, and most studies used screening questionnaires without final diagnostic outcomes, which all may have introduced bias into results, thereby underestimating the association with lockdown and warranting further research.

The present review has various strengths, including a comprehensive search with optimal subgroup and time-adjusted analysis that accounted for temporal trends in outcomes and explored inequalities between and within HICs. Publication bias was not present in any analyses. However, interpretation of our findings also requires consideration of limitations. For some outcomes, heterogeneity was concerning, a common challenge with aggregated data. To overcome this, we ran subgroup and stratified analysis where we had sufficient data. Discrimination, inequalities, health systems, lockdown restrictions and definitions vary between regions and countries and could contribute to high heterogeneity and limit comparability of data between studies[1]. However, we attempted to account for this variability by focusing on HICs, using random-effect estimates, and subgroup analysis by region and time period[8].

## Methods

We conducted a systematic review and meta-analysis of studies that explored the association between COVID-19 lockdowns and ABPOs in HICs. The review was registered on PROSPERO (CRD42022327448), where amendments and progress have been updated accordingly, and followed the relevant PRISMA guidance.

### Eligibility criteria

The intervention of interest was COVID-19 lockdowns, defined as those non-pharmaceutical, home-confinement and non-essential service closure interventions imposed by governments in 2020 with the aim of reducing the spread of COVID-19.

To be included in the review, studies had to report on at least one of the following ABPOs or outcome subclassifications (definitions in Supplementary Material 1).

ABPOs

Perinatal/neonatal outcomes:

- PTB
- Stillbirth
- LBW
- Neonatal mortality
- Neonatal (neonatal intensive care unit) admissions
- Hypoxic ischaemic encephalopathy
- Prolonged stay in hospital

Maternal outcomes:

- Caesarean section
- OASI
- Peripartum hysterectomy
- Sepsis (in puerperium)
- Prolonged stay in hospital
- Readmission to hospital
- Maternal mental health (depression or anxiety antenatally or postpartum)

The outcomes of this review were decided upon on the basis of current literature, expert advice, importance to patients and pragmatic considerations. Patients' perspectives and priorities were obtained from a women's reference group; the group was formed to consult on the COVID Maternity Equality Project study[169], where S.I. was the

principal investigator. Their insights and priorities informed which outcomes were included and explored in this review and research. Pragmatic considerations included data access and availability.

The population was restricted to those residing in HICs, as defined by the World Bank Classifications[170]. To be included, studies had to present a comparator cohort from before lockdowns or the pandemic in 2020. We included the following study types: observational studies, case–control studies, cohort studies and brief reports comparing outcomes before and during lockdowns. Studies had to be published between 1 January 2019 and 22 June 2023. Language requirements entailed that title and abstracts had to be written in English.

We excluded non-HIC populations or countries that did not enforce lockdowns, such as Sweden. We excluded studies that explored only COVID-19 infection in pregnancy and neonates. We also excluded qualitative studies, systematic reviews, narrative reviews, discussion articles, viewpoints/opinion articles and editorials.

### Search strategy and selection

Search terms were developed to include all key perinatal, neonatal and maternal health outcomes and word variants of COVID-19 lockdowns (Supplementary Material 2). We searched EMBASE, MEDLINE/PubMed and Web of Science. We also searched for preprints (MedRxiv) and unpublished reports; these were identified by co-authors through research networks. References of eligible studies were reviewed to identify additional eligible studies. Search results were exported to Endnote, where duplicates were removed, both automatically and manually. The results were then independently reviewed by two reviewers (H.N.S. and I.H.), who applied the eligibility criteria first to titles, then to abstracts and full texts. Reviewers then compared selections, and any disagreements were resolved by discussion between reviewers and a third author arbitrated (S.I.).

### Data extraction

Key characteristics of studies and data on outcomes and total births were extracted using a data extraction tool developed in Microsoft Excel version 16.85. Aggregated data stratified according to deprivation and ethnicity were also extracted. Multiple indicators of deprivation were used (Supplementary Material 3). Deprivation was an indicator variable, where data on maternal education, mean neighbourhood income and deprivation indices were pragmatically combined and harmonized to indicate high, medium or low deprivation (details available in Supplementary Material 3). Where data were missing, illegible or presented in the wrong format, we emailed authors to ask them to provide relevant aggregated data. If the authors did not respond, then the raw values were calculated on the basis of available information; however, in cases where information was insufficient for calculations, the study was excluded from meta-analysis. Risk of bias in each study was assessed and scored using the Newcastle–Ottawa risk of bias assessment tool for cohort studies[171]. After extraction, studies that had a cohort overlap were removed, where the smaller study was removed in favour of the larger study.

### Statistical analysis

Meta-analysis was conducted on outcomes that had more than three studies presenting relevant data, to calculate accurate and meaningful pooled estimates. For each outcome, we determined the random-effects estimate pooled risk ratio to account for heterogeneity across studies and populations, and 95% confidence intervals were calculated, as well as an $I^2$ value summarizing interstudy heterogeneity[172]. All statistical tests were two-tailed. Where heterogeneity was considerable (that is, >75%) and sufficient data were available, subgroup and stratified meta-analyses were conducted to explore potential sources of heterogeneity[172].

Due to the differing measures and implementation of lockdowns between countries, a high degree of spatial heterogeneity was expected and subgroup analyses were conducted according to continent[8]. Studies that reported data on a second defined lockdown period and/or a post-lockdown period were also meta-analysed as subgroup analysis. The reference period for both was the pre-pandemic epoch. This analysis attempted to explore whether the associations with the second lockdown varied greatly from associations with the first lockdown. To improve insight into lockdowns' association with distinct pathological processes, subclassifications of clinical outcomes were explored where data were available in supplementary analyses, for instance, spontaneous compared with iatrogenic (medically induced) PTB[8]. There are various subclassifications of outcomes included; these are listed with outcome definitions (Supplementary Material 1).

To explore inequalities within HICs, stratified meta-analysis according to deprivation and ethnicity was conducted where more than three studies presented sufficient data. Studies with stratified data were included, and subgroup pooled estimates were calculated and compared.

We also controlled for underlying temporal trends in outcomes when data were available. Failing to account for such trends can lead to spurious results[8,173]. Therefore, we conducted a meta-analysis of time-adjusted studies; we excluded studies without adjustment for underlying temporal trends using a quasi-experimental design[173,174]. The same pathway as the main analysis was followed for selection and data extraction. Two reviewers (B.G. and I.H.) independently screened studies previously identified as eligible for inclusion. Additional data were extracted from the studies, including total sample sizes, time-adjusted risk and odds ratios, and corresponding upper and lower confidence intervals. In papers where odds ratios were presented, they were converted to risk ratios using the formula described by Faber and colleagues[175]. Once data were extracted, an inverse-variance meta-analysis was conducted using the random-effects pooled estimate[172].

### Sensitivity analyses

Deprivation included several proxy variables to indicate an area's level of deprivation. To explore the association between area deprivation and ABPOs (when available) during lockdown more closely, a sensitivity analysis was conducted in which only studies reporting data stratified by an area deprivation composite index scores were included. Data harmonization of area deprivation index scores is outlined in Supplementary Material 3.

Funnel plots were constructed, and Egger's tests were run on outcomes that had data aggregated from more than ten studies to explore asymmetry and publication bias. For the Egger's test, a significance threshold of $P < 0.1$ was adopted. All analyses were conducted in Stata 17.0.

Among outcomes that had insufficient evidence to support an association with lockdown, we investigated the statistical power of samples to detect the pooled estimate and the minimum detectable effect.

### Role of funding source

The funder of the study is the National Institute of Health Research: School of Primary Care Research. The funder had no role in the study design, data collection and analysis, or presentation of the report.

## Conclusion

In conclusion, our systematic review and meta-analyses showed that, in HICs, a decrease in the likelihood of PTB was associated with lockdowns; however, this reduction was unequal among HICs and was apparent only in Europe and Australia. COVID-19 lockdowns were a natural experiment that imposed system and wider socio-environmental interventions. The uneven associations between ABPOs and lockdowns need to be explored further through mixed-methods investigation to assess the health determinants, societal structures and barriers to care that drive ABPOs[8,14]. Our findings provide evidence that an umbrella lockdown approach has unequal consequences for ABPOs; targeted policies and tailored support structures are warranted to achieve and promote maternal and neonatal health[14].

## Reporting summary

Further information on research design is available in the Nature Portfolio Reporting Summary linked to this article.

## Data availability

All data used in the meta-analysis were generated from deidentified, aggregated data extracted from studies included in the meta-analysis or were shared with authors upon request. This study makes use of publicly available data and data already published in primary research studies. In terms of a minimum dataset, all aggregated data used in meta-analysis are listed on corresponding forest plots. Individuals seeking access to unpublished data require relevant permissions from individual study authors.

## Code availability

All code used in the meta-analysis was based on predeveloped code available in Stata 17.0 meta-analysis manuals. All code is available upon request.

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

## Acknowledgements

We thank the funder of this project, the National Institute of Health Research: School of Primary Care Research (grant number SPCR C007) and Tommy's National Preterm Birth Research Centre.

## Author contributions

I.H., S.I. and D.Z. conceptualized the study aims, study design, inclusion and exclusion criteria and data curation. I.H., H.N.S. and B.Y.G. conducted data searches and independent screening of studies. I.H. and H.N.S. conducted data extraction. I.H. was responsible for data organization and analysis. S.I. and D.Z. participated in and advised study selection, data analysis and visualization. I.H., J.V.B. and L.B.-O. conceptualized and provided input on secondary nested time-adjusted analysis, including providing input on study selection, data organization, analysis and visualization. I.H. wrote all drafts of the paper with input from J.J., S.I., D.Z., J.V.B. and L.B.-O. All authors had full access to all the data in the study and agreed to submit for publication. I.H., S.I., D.Z. and L.B.-O. have accessed and verified the data included in meta-analysis.

## Competing interests

J.J. was a co-author of the Royal College of Obstetrics and Gynecology (RCOG) guidelines, which determined changes to maternity care during the COVID-19 pandemic. The other authors declare no competing interests.

## Additional information

**Correspondence and requests for materials** should be addressed to Iona Hindes.

Iona Hindes [1] ✉, Hawa Nuralhuda Sarwar[1], Benjamin Y. Gravesteijn[2], Jennifer Jardine [1], Lizbeth Burgos-Ochoa[3,4], Jasper V. Been [3,5,6], Dominik Zenner [7,8] & Stamatina Iliodromiti[1,8]

[1]Women's Health Research Unit, Centre for Public Health and Policy, Wolfson Institute of Population Health, Queen Mary University London, London, UK. [2]Amsterdam Reproduction and Development, Amsterdam University Medical Centre, Amsterdam, the Netherlands. [3]Department of Obstetrics and Gynaecology, Erasmus MC, Sophia Children's Hospital, University Medical Centre Rotterdam, Rotterdam, the Netherlands. [4]Department of Methodology and Statistics, Tilburg University, Tilburg, the Netherlands. [5]Division of Neonatology, Department of Neonatal and Paediatric Intensive Care, Erasmus MC Sophia Children's Hospital, University Medical Centre Rotterdam, Rotterdam, the Netherlands. [6]Department of Public Health, Erasmus MC, University Medical Centre Rotterdam, Rotterdam, the Netherlands. [7]Global Public Health Unit, Centre for Public Health and Policy, Wolfson Institute of Population Health, Queen Mary University London, London, UK. [8]These authors jointly supervised this work: Dominik Zenner, Stamatina Iliodromiti. ✉e-mail: i.hindes@qmul.ac.uk

# Reporting Summary

## Statistics

For all statistical analyses, confirm that the following items are present in the figure legend, table legend, main text, or Methods section.

| n/a | Confirmed | |
|---|---|---|
| ☐ | ☒ | The exact sample size (*n*) for each experimental group/condition, given as a discrete number and unit of measurement |
| ☐ | ☒ | A statement on whether measurements were taken from distinct samples or whether the same sample was measured repeatedly |
| ☐ | ☒ | The statistical test(s) used AND whether they are one- or two-sided<br>*Only common tests should be described solely by name; describe more complex techniques in the Methods section.* |
| ☐ | ☒ | A description of all covariates tested |
| ☐ | ☒ | A description of any assumptions or corrections, such as tests of normality and adjustment for multiple comparisons |
| ☐ | ☒ | A full description of the statistical parameters including central tendency (e.g. means) or other basic estimates (e.g. regression coefficient) AND variation (e.g. standard deviation) or associated estimates of uncertainty (e.g. confidence intervals) |
| ☐ | ☒ | For null hypothesis testing, the test statistic (e.g. *F*, *t*, *r*) with confidence intervals, effect sizes, degrees of freedom and *P* value noted<br>*Give P values as exact values whenever suitable.* |
| ☒ | ☐ | For Bayesian analysis, information on the choice of priors and Markov chain Monte Carlo settings |
| ☒ | ☐ | For hierarchical and complex designs, identification of the appropriate level for tests and full reporting of outcomes |
| ☒ | ☐ | Estimates of effect sizes (e.g. Cohen's *d*, Pearson's *r*), indicating how they were calculated |

*Our web collection on statistics for biologists contains articles on many of the points above.*

## Software and code

Policy information about availability of computer code

| Data collection | Microsoft Excel version 16.85 |
|---|---|
| Data analysis | Stata 17.0 |

For manuscripts utilizing custom algorithms or software that are central to the research but not yet described in published literature, software must be made available to editors and reviewers. We strongly encourage code deposition in a community repository (e.g. GitHub). See the Nature Portfolio guidelines for submitting code & software for further information.

## Data

Policy information about availability of data

All manuscripts must include a data availability statement. This statement should provide the following information, where applicable:

- Accession codes, unique identifiers, or web links for publicly available datasets
- A description of any restrictions on data availability
- For clinical datasets or third party data, please ensure that the statement adheres to our policy

All data used in the meta-analysis, was generated from de-identified, aggregated data extracted from studies included in meta-analysis or was shared with authors upon request. This study makes use of publicly available data and data already published in primary research studies.
In terms of a minimum dataset, all aggregated data used in meta-analysis are listed on corresponding forest plots, thus allowing for ease of interpretation, verification, and transparency to readers.

Individuals seeking access to unpublished data require relevant permissions from individual study authors.

Databases were searched for original research publications, databases searched included: EMBASE, MEDLINE/PubMed, Web of Science and MedRxiv. The search term used in such databases can be found in Appendix B.

## Research involving human participants, their data, or biological material

Policy information about studies with [human participants or human data](). See also policy information about [sex, gender (identity/presentation), and sexual orientation]() and [race, ethnicity and racism]().

| | |
|---|---|
| Reporting on sex and gender | *Use the terms sex (biological attribute) and gender (shaped by social and cultural circumstances) carefully in order to avoid confusing both terms. Indicate if findings apply to only one sex or gender; describe whether sex and gender were considered in study design; whether sex and/or gender was determined based on self-reporting or assigned and methods used.*<br>*Provide in the source data disaggregated sex and gender data, where this information has been collected, and if consent has been obtained for sharing of individual-level data; provide overall numbers in this Reporting Summary. Please state if this information has not been collected.*<br>*Report sex- and gender-based analyses where performed, justify reasons for lack of sex- and gender-based analysis.* |
| Reporting on race, ethnicity, or other socially relevant groupings | *Please specify the socially constructed or socially relevant categorization variable(s) used in your manuscript and explain why they were used. Please note that such variables should not be used as proxies for other socially constructed/relevant variables (for example, race or ethnicity should not be used as a proxy for socioeconomic status).*<br>*Provide clear definitions of the relevant terms used, how they were provided (by the participants/respondents, the researchers, or third parties), and the method(s) used to classify people into the different categories (e.g. self-report, census or administrative data, social media data, etc.)*<br>*Please provide details about how you controlled for confounding variables in your analyses.* |
| Population characteristics | *Describe the covariate-relevant population characteristics of the human research participants (e.g. age, genotypic information, past and current diagnosis and treatment categories). If you filled out the behavioural & social sciences study design questions and have nothing to add here, write "See above."* |
| Recruitment | *Describe how participants were recruited. Outline any potential self-selection bias or other biases that may be present and how these are likely to impact results.* |
| Ethics oversight | *Identify the organization(s) that approved the study protocol.* |

Note that full information on the approval of the study protocol must also be provided in the manuscript.

# Field-specific reporting

Please select the one below that is the best fit for your research. If you are not sure, read the appropriate sections before making your selection.

☒ Life sciences　　☐ Behavioural & social sciences　　☐ Ecological, evolutionary & environmental sciences

For a reference copy of the document with all sections, see [nature.com/documents/nr-reporting-summary-flat.pdf]()

# Life sciences study design

All studies must disclose on these points even when the disclosure is negative.

| | |
|---|---|
| Sample size | This study was a systematic review, therefore, no sample size calculations were required. Sample size was determined based on availability of data and studies. |
| Data exclusions | Data duplicates were excluded or those which did not match the eligibility criteria of the review. |
| Replication | All details of the search strategy, study selection, data extraction and statistical analyses have been included to enable replication of the search and analysis. |
| Randomization | This study was a systematic review and meta-analysis using publicly available, aggregated data. Individual data was not available, thus randomization was not possible or necessary. |
| Blinding | This study was a systematic review and meta-analysis using publicly available, aggregated data. Individual data was not available, thus blinding was not possible or necessary. |

# Reporting for specific materials, systems and methods

We require information from authors about some types of materials, experimental systems and methods used in many studies. Here, indicate whether each material, system or method listed is relevant to your study. If you are not sure if a list item applies to your research, read the appropriate section before selecting a response.

## Materials & experimental systems

| n/a | Involved in the study |
|---|---|
| ☒ ☐ | Antibodies |
| ☒ ☐ | Eukaryotic cell lines |
| ☒ ☐ | Palaeontology and archaeology |
| ☒ ☐ | Animals and other organisms |
| ☒ ☐ | Clinical data |
| ☒ ☐ | Dual use research of concern |
| ☒ ☐ | Plants |

## Methods

| n/a | Involved in the study |
|---|---|
| ☒ ☐ | ChIP-seq |
| ☒ ☐ | Flow cytometry |
| ☒ ☐ | MRI-based neuroimaging |

# Plants

Seed stocks
*Report on the source of all seed stocks or other plant material used. If applicable, state the seed stock centre and catalogue number. If plant specimens were collected from the field, describe the collection location, date and sampling procedures.*

Novel plant genotypes
*Describe the methods by which all novel plant genotypes were produced. This includes those generated by transgenic approaches, gene editing, chemical/radiation-based mutagenesis and hybridization. For transgenic lines, describe the transformation method, the number of independent lines analyzed and the generation upon which experiments were performed. For gene-edited lines, describe the editor used, the endogenous sequence targeted for editing, the targeting guide RNA sequence (if applicable) and how the editor was applied.*

Authentication
*Describe any authentication procedures for each seed stock used or novel genotype generated. Describe any experiments used to assess the effect of a mutation and, where applicable, how potential secondary effects (e.g. second site T-DNA insertions, mosiacism, off-target gene editing) were examined.*

