## [Peer Review File · Nature Human Behaviour]

A systematic review and meta-analysis of the association of COVID-19 lockdowns with adverse birth and pregnancy outcomes in 28 high-income countries

Corresponding Author: Ms Iona Hides

Version 0:

Decision Letter:

16th May 2024

Dear Ms Hides,

Thank you once again for your manuscript, entitled "COVID-19 lockdowns' impact on birth and pregnancy, and associated inequalities, within high-income countries: a systematic review and meta-analysis.," and for your patience during the peer review process. Once again I sincerely apologise for the delay in our decision.

Your manuscript has now been evaluated by 2 reviewers, whose comments are included at the end of this letter. (As you know, a third reviewer agreed to review the work, but has been unable to submit their review; if we do hear from them we will forward their comments on to you.) Although the reviewers find your work to be of interest, they also raise some important concerns. We are interested in the possibility of publishing your study in Nature Human Behaviour, but would like to consider your response to these concerns in the form of a revised manuscript before we make a decision on publication.

Your revised manuscript must comply fully with our editorial policies and formatting requirements. Failure to do so will result in your manuscript being returned to you, which will delay its consideration. To assist you in this process, I have attached a checklist that lists all of our requirements. **In particular, please closely follow our requirements for statistical reporting and treatment of null results when revising your manuscript.** If you have any questions about any of our policies or formatting, please don't hesitate to contact me.

In sum, we invite you to revise your manuscript taking into account all reviewer and editor comments. We are committed to providing a fair and constructive peer-review process. Do not hesitate to contact us if there are specific requests from the reviewers that you believe are technically impossible or unlikely to yield a meaningful outcome.

We hope to receive your revised manuscript within two months. I would be grateful if you could contact us as soon as possible if you foresee difficulties with meeting this target resubmission date.

- Include a "Response to the editors and reviewers" document detailing, point-by-point, how you addressed each editor and referee comment. If no action was taken to address a point, you must provide a compelling argument. When formatting this document, please respond to each reviewer comment individually, including the full text of the reviewer comment verbatim followed by your response to the individual point. This response will be used by the editors to evaluate your revision and sent back to the reviewers along with the revised manuscript.
- Highlight all changes made to your manuscript or provide us with a version that tracks changes.

Link Redacted

We look forward to seeing the revised manuscript and thank you for the opportunity to review your work. Please do not hesitate to contact me if you have any questions or would like to discuss these revisions further.

Sincerely,

Reviewer expertise:

Reviewer #1: Epidemiology, newborn outcomes

Reviewer #2: Systematic reviews and meta analyses, epidemiology, birth outcomes

REVIEWER COMMENTS:

Reviewer #1:

Remarks to the Author:

This analysis focuses on an important topic as countries move forward post-COVID and aim to plan better for the future. I had a few queries/suggestions:

1. DEPRIVATION: This was mentioned as an important factor but I struggled to understand how this was analyzed as a variable (in contrast to ethnicity which is more traditional and well-understood). In the Methods, it says that, "Multiple indicators of deprivation were used (Appendix C)." In the appendix, the different definitions applied for deprivation are listed in addition to a deprivation index but how this index was calculated was not clear. Deprivation as a construct could only be examined in 14 of the 132 studies overall, correct? Perhaps reporting separately about each factor (e.g. education, income, SES position, etc.) from the subset analyzed would be more clear because general conclusions based upon "deprivation" groupings were not convincing. At quick glance, even determining low, med, high deprivation based upon income in a standard way for a population drawn from Chicago (Illinois) versus Tennessee would be difficult.
2. SGA: Curious if any of the hospital-based studies offered meaningful insight about the role or contribution of SGA related to the LBW rates. I noted that both PTB and LBW were analyzed/presented as outcomes and SGA was in the appendix but not mentioned.
3. PTB: The appendix showed that there were sub-classifications based upon extreme or very PTB and just wondering if the PTB findings were consistent or differed from those focusing on these more medically vulnerable subgroups.

Lastly, this may be a comment for further investigation but it seems to me that the impact of a lockdown enacted as PHI in a country would operate to limit SUPPLY or DEMAND for healthcare services. You describe the intervention as two key components (1-home-confinement and 2-non-essential service closure interventions) imposed by governments but the impact on ABPOs may vary with whether national or local policies included either or both of the components. In other words, was access for women restricted or were facilities simply not open/operating? This may be a key difference because in LMIC or resource-limited settings, both delays served as formidable barriers. Assessing lockdowns broadly or more universally as a blanket intervention may have been necessary perhaps due to data limitations and this could be mentioned or addressed in the discussion.

Reviewer #2:

Remarks to the Author:

This was an interesting systematic review that included a large number of papers and concerned an interesting topic, but I wondered at the strategy of focusing on differences in maternal and birth outcomes within high income countries only, where the variation in COVID-19 lockdown impacts would be most difficult to demonstrate, given the existence of well-established and (generally) high-quality national health care systems.

The rationale for the study was clear but the discussion was not clearly linked to the rationale. In the introduction, the importance of identifying inequitable outcomes among SE groups with countries was emphasised. The fact that the analysis was not able to identify the expected differences should have been fully discussed and explained in the discussion section, situating this finding within the broader literature on health inequality.

Specific comments:

Outcomes according to 'ethnicity' and 'deprivation' were the subject of sub-analyses, but these results were not discussed. As this might be considered by many to be an unexpected finding, some explanation is required beyond the discussion of heterogeneity in the limitations section.

The methods section was well set out and much of the additional information required was available in supplementary files. I did wonder how case-control studies were handled in the analyses. By definition, case-control study participants are selected on the basis of their outcome status, so combining data from case-control studies and cohort studies to develop a pooled estimate does not seem intuitive.

The characteristics of the studies should be summarised in the Results section. e.g., how many countries were included and the relative proportions of those countries who may have been over-represented, the types of study designs used, etc.

The tables of included studies (appendix D) should include a column for study design (cohort, case-control, case series, etc.)

Minor issues:

Line 96 "In early 2020, the global spread of SARS-CoV-2 (COVID-19) virus prompted governments..." should be rephrased. COVID-19 is the disease caused by SARS-CoV-2, and does not describe the virus itself. e.g., "In early 2020, the global spread of the SARS-CoV-2 virus (leading to the infection, COVID-19) prompted governments..."

Line 125-6, "Lockdowns impacted ABPOs differently in HICs compared to LMICs, and within HICs lockdowns impact varied according to region, time-period, and pre-existing inequalities." This might need to be re-phrased as it currently reads that the authors are claiming there is no health or social gradient in lower income countries.

Line 139 "...with the aim of reducing the spread of COVID-19 infection." COVID-19 is infection with SARS-CoV-2, so the word 'infection' is redundant in this sentence

In several places, the authors refer to data as singular rather than plural (e.g., the text should read 'data were' rather than 'data was')

Line 242-3 "All analyses were conducted in 243 STATA 17.0." Stata should be written in title case not capitals - i.e., Stata is not an acronym

Line 393 the word 'Covid-19' should be replaced by 'COVID-19'.

Version 1:

Decision Letter:

Our ref: NATHUMBEHAV-23103508A

9th August 2024

Dear Dr. Hinde,

Thank you for submitting your revised manuscript "The impact of COVID-19 lockdowns on adverse birth and pregnancy outcomes, and associated inequalities, in high-income countries: a systematic review." (NATHUMBEHAV-23103508A). Unfortunately the original referees were unable to re-review; however it has now been seen by a third reviewer whose expertise overlaps with the reviewers from the previous round, and their comments are below. As you can see, the reviewer finds that the paper has improved in revision, but raises a number of important points. We will be happy in principle to publish it in Nature Human Behaviour, pending minor revisions to satisfy the referee's final requests and to comply with our editorial and formatting guidelines. In particular, we will ask that you remove all causal language and interpretation of null findings (or provide Bayes Factors to interpret null results).

We are now performing detailed checks on your paper and will send you a checklist detailing our editorial and formatting requirements within two weeks. Please do not upload the final materials and make any revisions until you receive this additional information from us.

Sincerely,

[Redacted signature]

[Redacted signature]

Nature Human Behaviour

Reviewer #3 (Remarks to the Author):

The authors, in response to the reviewer comments in their revised manuscript, have sufficiently expanded on their explanations of the methods, results and discussions.

The crux of the manuscript, abstract and overall conclusion of the authors is that lockdowns during COVID-19 pandemic were associated with reduced PTB, which was more so in Europe and Australia, and in white ethnicity and low deprivation groups based on their subgroup analysis. However only in the regional subgroup analysis by continent was there significant subgroup differences in PTB (Figure 2, Panel B $p=0.000$), ethnicity group Figure 2, Panel C ($p=0.268$), and deprivation levels Figure 2, Panel D ($p=0.503$).

The authors set out to explore reasons for heterogeneity with their subgroup analysis, and substantial to moderate heterogeneity still remained within each subgroup of their analysis as high as 92% in Asian (for Panel B), and 63% in high deprivation (Panel D). It doesn't look like the subgroup analysis has explained heterogeneity.

A correct interpretation of these figures would be that "The test for subgroup differences suggests that there is no statistically significant subgroup effect for ethnicity ($p=0.268$) or deprivation levels ($p=0.503$) on PTB".

I would therefore suggest the authors tone down interpretations of their findings with regards to these 2 stratifications for PTB (ethnicity and deprivation levels). Otherwise this was a very interesting study which I enjoyed reading and would benefit from publication.

Version 2:

Decision Letter:

Dear Ms Hindes,

We are pleased to inform you that your Article "A systematic review and meta-analysis of the association of COVID-19 lockdowns with adverse birth and pregnancy outcomes in 28 high-income countries", has now been accepted for publication in Nature Human Behaviour.

With best regards,

[REDACTED]

[REDACTED]

Nature Human Behaviour

P.S. Click on the following link if you would like to recommend Nature Human Behaviour to your librarian
<http://www.nature.com/subscriptions/recommend.html#forms>

** Visit the Springer Nature Editorial and Publishing website at http://editorial-jobs.springernature.com?utm_source=ejp_NHumB_email&utm_medium=ejp_NHumB_email&utm_campaign=ejp_NHumB for more information about our career opportunities. If you have any questions please click [here](mailto:editorial.publishing.jobs@springernature.com). **

Response to the Editors and Reviewers

We would like to thank the editors and reviewers for taking the time to review our manuscript. We have now responded to their comments point by point and edited our manuscript accordingly. Please see our responses to specific comments in the indented sections below.

REVIEWER COMMENTS:

Reviewer #1:

Remarks to the Author:

This analysis focuses on an important topic as countries move forward post-COVID and aim to plan better for the future. I had a few queries/suggestions:

Reviewer #1, comment: 1. DEPRIVATION: *This was mentioned as an important factor but I struggled to understand how this was analyzed as a variable (in contrast to ethnicity which is more traditional and well-understood). In the Methods, it says that, "Multiple indicators of deprivation were used (Appendix C)." In the appendix, the different definitions applied for deprivation are listed in addition to a deprivation index but how this index was calculated was not clear. Deprivation as a construct could only be examined in 14 of the 132 studies overall, correct? Perhaps reporting separately about each factor (e.g. education, income, SES position, etc.) from the subset analyzed would be more clear because general conclusions based upon "deprivation" groupings were not convincing. At quick glance, even determining low, med, high deprivation based upon income in an standard way for a population drawn from Chicago (Illinois) versus Tennessee would be difficult.*

Response to Reviewer #1 comment 1:

We thank the reviewer for the positive remarks and support of the research. We appreciate the feedback and hope that in the following section we address their concerns regarding the deprivation analysis.

Deprivation is commonly defined as a composite of several indicators of individual or small-area level (dis)advantage. In this study, there is substantial variation between studies in their definition of deprivation and variables used to indicate deprivation. Definitions, as the reviewer makes clear, depend on local and national context.

With this in mind, we have taken a pragmatic approach to synthesis. Most studies who explored deprivation, reported data stratified according to area deprivation indices, such as Index of Multiple Deprivation, while other studies reported data stratified according to individual socioeconomic factors, such as maternal education. Our variable of deprivation included both types of variables and constituted an indicator rather than a standardized index. As some studies did not report deprivation indices, we included proxy variables which could reasonably be used to estimate deprivation, such as maternal education or mean income level. We pragmatically allocated each of the groups of these variables to indicate high,

medium, or low deprivation. Data was reported differently between studies, and therefore we attempted to harmonise data in the following way: first all eligible studies were reviewed to identify any variables on deprivation, or which could act as a proxy for deprivation, e.g. income or education. Each group from each variable, and its relevant data, was then allocated to low, medium, or high deprivation. For instance, low maternal education (less than high school diploma or equivalent) was categorized as high deprivation. The process of how the deprivation variable was constructed has been outlined in more detail in Appendix C and integrated within the manuscript. We have outlined below, which text has been added to the publication according to the reviewer's comments.

Regarding the reviewer's comment of conducting a separate subset analysis: 14 studies reported on deprivation or ethnicity, only 7 studies presented stratified results by deprivation indices or deprivation proxy variables. 4 of these 7 studies reported neighbourhood deprivation index or disadvantage index (based on a composite of socioeconomic factors). 1 study reported maternal education level at time of birth, and 2 reported mean neighbourhood income level. In response to the reviewer's comment, we have added a sensitivity analysis, in which all studies using proxy variables of deprivation have been removed, and only studies which used a deprivation composite index measure were included. We analysed 4 studies with 1,484,858 livebirths. We found that, when only studies including area deprivation composite indices were included in analysis compared to the main analysis, the direction and magnitude of effect decreased slightly for low deprivation (pooled RR:0.97, 95%CI:0.93-1.01), increased for medium deprivation (pooled RR:0.91, 95%CI:0.85-0.97), and remained similar for high deprivation (pooled RR:0.97, 95%CI:0.92-1.02). The results of this analysis have been added to the manuscript as a post-hoc sensitivity analysis and are included below.

As the reviewer has noted, the data used in this analysis is not uniform and risks the misclassification of individuals, this limits the accuracy of estimates and likely underestimates the magnitude of inequalities through non-differential misclassification. The implications of this analysis and the limitations of standardized deprivation measurements have been added to the discussion. Nevertheless, we have taken a pragmatic approach to overcome this issue and believe this analysis is highly relevant as it indicates how deprivation and socio-economic-related factors influence in the impact of lockdown on the outcomes under study. Most importantly this research indicates lockdowns affected inequalities in preterm birth. The reduction in preterm birth during lockdowns was greatest among those living in low and medium deprivation settings, and White individuals. The effect of lockdown on inequalities is likely underestimated in our analysis, due to the data limitations we have described, and requires further investigation to be clearly understood and prevented in the future. We have added more details about the limitations of this approach, as highlighted by the reviewer, and sensitivity analyses, as suggested by the reviewer, to the paper.

Below we have outlined what text has been added to which sections of the paper.

The following text was added to Methods:

“Deprivation was an indicator variable, where data on maternal education, mean neighbourhood income, and deprivation indices were pragmatically combined and harmonised to indicate high, medium, or low deprivation (details available in Appendix C).”

The following text was added to Statistical Analysis:

“Sensitivity Analyses:

Deprivation included several proxy variables to indicate an area’s level of deprivation. To explore the association between area deprivation and ABPOs (when available) during lockdown more closely, a sensitivity analysis was conducted in which only studies reporting data stratified by an area deprivation composite index scores were included. Data harmonisation of area deprivation index scores is outlined in Appendix C.”

The following text was added to the Results:

“Sensitivity Analyses:

When only studies which stratified data according to area deprivation composite indices were included in the analysis of PTB during lockdowns, the direction and magnitude of effect decreased slightly for low deprivation (RR0.97, 95%CI:0.93-1.01), increased for medium deprivation (RR0.91, 95%CI:0.85-0.97), and remained similar for high deprivation (RR0.97, 95%CI:0.92-1.02) (Figure 6).

NOTE: Weights and between-subgroup heterogeneity test are from random-effects model

(Figure 6. Forest plot of risk ratios of preterm birth over lockdown compared to pre-lockdown, stratified by deprivation level (only area deprivation level composite indices included))”

The following text was added to the Discussion:

“Investigation of inequalities in ABPOs between ethnicity and deprivation groups was limited due to data availability and heterogeneity between studies and settings. Very few studies reported data stratified according to ethnicity and deprivation, and fewer used the same ethnicity groups or deprivation levels. This is understandable as ethnicity distribution varies between settings, and deprivation indicators such as maternal education and neighbourhood income are contextually specific. We addressed this limitation pragmatically by synthesizing available data on deprivation or by using proxy variables for deprivation, and harmonising composite and proxy variables to obtain aggregated estimates of deprivation.

In our analysis, only the white ethnicity group showed evidence of a reduction in PTB associated with the lockdown period. Other, Hispanic, and Asian ethnicity groups indicated a decreasing trend in PTB over the lockdown period, however there was insufficient evidence to support an association. The Black ethnicity group indicated no evidence to support a change in PTB rates associated with the lockdown period. Hence, it appears that the association between lockdown restrictions and PTB differed according to ethnicity, potentially affecting the magnitude of inequalities among ethnicity groups in HICs.

In terms of deprivation, stratified analysis showed that only the least deprived groups had a decrease in PTB associated with the lockdown period. PTB in medium and high deprivation groups did not materially change from pre-lockdown to lockdown period. Area level estimates of deprivation have an inherent limitation of misclassification of individuals, for instance those who live in areas of high deprivation but are not themselves deprived; this non-differential misclassification blunts the gradient of observed differences in deprivation. When these estimates are combined, this is compounded, underestimating the association which biases towards the null. Our results indicate that lockdowns were associated with unequal impacts on PTB according to socio-economic deprivation and ethnicity, however, the magnitude of the effect may be underestimated and analysis using individual deprivation indices is warranted.”

The following text was added to the Appendices:

“Appendix C

Deprivation Indicator Variable

Some studies reported data stratified by patient’s area deprivation index. An area deprivation index uses a variety of socio-economic factors to estimate the level of socioeconomic deprivation in an area where a person lives, the area is then allocated

to one of five deprivation quintiles ranging from low to high. However, some studies did not report data according to a deprivation index, but instead reported data based on individual socioeconomic factors, such as maternal education level, or an area's average income. Some of these variables could be used as a proxy for deprivation.

Studies which presented data stratified according to deprivation index or a relevant proxy variable for deprivation were identified. Individual studies were not assigned to deprivation levels, their groups of stratified data were. For each of these studies presenting relevant data, the stratified data was extracted and allocated to low, medium, and high deprivation sub-groups. The process of harmonising data to deprivation levels is outlined in the table below.

The table presents each study included in the meta-analysis, which presented data stratified according to a deprivation index or a proxy variable. Each row is allocated to a single study which presented data stratified according to deprivation or a relevant proxy variable. In the first columns of the table the core details of the study are identified. In the following columns we identify the variable of deprivation which the study used (either a deprivation index or a proxy variable), the source of the variable, if a deprivation index was used then we outline what socioeconomic factors are included in the index. The subsequent columns outline how the groups of the deprivation variable are defined by the data source, followed by how the individual study presented data (some studies aggregated groups or quintiles to indicate higher / lower deprivation). In the final column, we indicate what level of deprivation (low, medium, high) we allocated to each group of the variable, in each study.

First author last name and year of publication	Variable used to estimate deprivation	Data source of variable	Definition of variable	If an index, sociodemographic factors included in index calculation	Group definitions of variable according to the data source	Group as defined in publication, by which data was presented in the paper	Level of deprivation group allocated to in our analysis
Harvey 2021	Maternal education	Tennessee birth records	Education level of mother	N/a	less than high school	less than high school	high deprivation
					high school/general education diploma	high school/general education diploma	medium deprivation
					some college/associate degree	some college/associate degree	low deprivation
					college degree	college degree	
Shah 2021	Neighbourhood income quintile	Statistics Canada postal code conversion file	Neighbourhood area level income	N/a	quintile 1 (poorest)	quintile 1 (poorest)	high deprivation
					quintile 2	quintile 2	
					quintile 3	quintile 3	medium deprivation
					quintile 4	quintile 4	low deprivation
					quintile 5 (richest)	quintile 5 (richest)	

Aboulatta 2023	Mean household income	Manitoba Population Research Data Repository	Census data for income quintiles based on ranges of mean household income, and grouped into five categories with each quintile assigned to approximately 20% of the population (quintile 1 (mean income=\$C17 910) to quintile 5 (mean income=\$46 230)).	N/a	lower income (individuals in the lowest and second lowest median neighbourhood income quintile)	lower income (individuals in the lowest and second lowest median neighbourhood income quintile)	high deprivation
------------------------------	---	--	------------	--	--	-------------------------

					higher income (individuals residing in the neighbourhoods with the three highest median neighbourhood income quintiles)	higher income (individuals residing in the neighbourhoods with the three highest median neighbourhood income quintiles)	low deprivation
Been 2020	Neighbourhood socioeconomic status	Netherlands Institute for Social Research	Neighbourhood area level deprivation	Mean household income, proportion of population with low income, proportion of population with low education level, proportion of population without paid work.	quintile 1 (poorest)	low (<p20)	high deprivation
					quintile 2	medium (p20-80)	medium deprivation
					quintile 3		
					quintile 4	high (>=p80)	low deprivation
					quintile 5 (richest)		
Fisher 2022	Neighbourhood deprivation index	United States Census Bureau. American community survey 5-year data (2009-2019).	Neighbourhood area level deprivation	Dependency, educational attainment, unemployment, poverty, per capita	first	lower three quartiles	low deprivation
					second		
					third		

		2020. Available at: https://www.census.gov/data/developers/data-sets/acs-5year.html .		income, and crowded housing	fourth	highest quartile	high deprivation
Guro-Urganci 2022	Neighbourhood deprivation index	2019 Index of Multiple Deprivation, Available at: https://www.gov.uk/government/statistics/english-indices-of-deprivation-2019)	2019 Area Index of Multiple Deprivation	Income, education, employment, crime, and living environment in an individual's area of residence	quintile 1 (least deprived)	less deprived	low deprivation
					quintile 2		
					quintile 3		
					quintile 4	more deprived	high deprivation
quintile 5 (most deprived)							
Lemon 2021	Neighbourhood deprivation index	Department of Medicine, School of Medicine and Public Health, University of Wisconsin. Neighborhood Atlas. 2015. Available at: https://www.neighborhoodatlas.medicine.wisc.edu/ .	Area deprivation index	Income, education, employment, and housing quality	ADI tertile 1 (1-52)	ADI tertile 1 (1-52)	low deprivation
					ADI tertile 2 (53-75)	ADI tertile 2 (53-75)	medium deprivation
					ADI tertile 3 (75 - 100)	ADI tertile 3 (75 - 100)	high deprivation

Reviewer #1. comment: 2. SGA: Curious if any of the hospital-based studies offered meaningful insight about the role or contribution of SGA related to the LBW rates. I noted that both PTB and LBW were analyzed/presented as outcomes and SGA was in the appendix but not mentioned.

Response to Reviewer #1 comment 2:

SGA was included in the supplementary material and appeared to decrease by a similar margin as LBW. As PTB decreased by a statistically significant amount, and both LBW and SGA did not, the contribution of SGA to the (non-statistically significant) reduction in LBW is likely negligible. The outcomes of SGA and LBW require a longer amount of time to occur than PTB does; this could explain why SGA and LBW indicated a decreasing trend over the lockdown period, but there was insufficient time for these outcomes to develop for there to be enough evidence of a substantial decrease.

Of the publications we included, only n=3 studies included both SGA and LBW in their reported outcomes. The details of the studies are reported below, along with the rates of SGA and LBW in the lockdown and pre-lockdown comparison period. The rates of SGA were close to the rates of LBW.

First author last name & year of publication		Pre-Lockdown	Lockdown	Average of Pre-lockdown and lockdown rates
Amadori 2021	SGA (rate per 1000 live births)	4.4	5.1	4.8
	LBW (rate per 1000 live births)	6.5	6.3	6.4
Maki 2023	SGA (rate per 1000 live births)	7.3	5.2	6.25
	LBW (rate per 1000 live births)	7.9	6.5	7.2
Rolnik 2021	SGA (rate per 1000 live births)	10.4	10.1	10.25

	LBW (rate per 1000 live births)	10.2	8.8	9.5
--	---------------------------------	------	-----	-----

Reviewer #1, comment: 3. *PTB: The appendix showed that there were sub-classifications based upon extreme or very PTB and just wondering if the PTB findings were consistent or differed from those focusing on these more medically vulnerable subgroups.*

Response to Reviewer #1 comment 3:

These outcomes were also included in the supplementary material. The point estimates of the impact of lockdowns on PTB were similar across gestational ages, including extreme and very PTB. However, it is likely this decrease only obtained statistical significance in moderate-to-late PTB and overall PTB, as extreme and very PTB are rare outcomes with much smaller incidence rates. This has been outlined more clearly in the discussion and supplementary analysis, details of the text added to the paper is included below:

The following text was added to the Discussion:

“We showed that lockdowns were associated mainly with decreases in spontaneous PTB and were similar across gestational ages (Supplementary Material: Figure 3: Panel C).”

The following text was added to the Supplementary Material:

“Meta-analysis of extreme PTB included 28 studies, which indicated a decreasing trend during the lockdown period compared to pre-lockdown (RR0.97[95%CI0.91-1.03]) (Figure 3: Panel A). Very PTB, reported by 28 studies, also suggested a possible decrease during the lockdown period (RR0.93[95%CI0.83-1.04]) (Figure 3: Panel B). Moderate to late PTB was reported by 21 studies, and had credible evidence of a decrease (6%) over the lockdown period (RR0.94[95%CI0.92-0.97]) (Figure 3: Panel C).”

Reviewer #1, comment: 4 *Lastly, this may be a comment for further investigation but it seems to me that the impact of a lockdown enacted as PHI in a country would operate to limit SUPPLY or DEMAND for healthcare services. You describe the intervention as two key components (1-home-confinement and 2-non-essential service closure interventions) imposed by governments but the impact on ABPOs may vary with whether national or local policies*

included either or both of the components. In other words, was access for women restricted or were facilities simply not open/operating? This may be a key difference because in LMIC or resource-limited settings, both delays served as formidable barriers. Assessing lockdowns broadly or more universally as a blanket intervention may have been necessary perhaps due to data limitations and this could be mentioned or addressed in the discussion.

Response to Reviewer #1 comment 4:

We agree with the reviewer that lockdowns likely did impact both the supply and demand for health care services. However, in HICs most maternity services remained open throughout lockdowns. This is a key distinction to LMICs where services may have closed because of restrictions. Therefore, the supply side of maternity care was not limited to the same degree as in LMICs. This was part of the rationale to focus on HICs as there is evidence, referenced in the manuscript, that maternity services remained open and operated throughout lockdowns in HICs. However, access to care was affected in a multitude of ways by restrictions, which possibly impacted the delivery of services as well as individuals' likelihood to seek care. More in-depth country-level data would be required from studies to explore these factors in this review. Given the HIC setting and limits on data availability, there are limitations on what we can elicit regarding these demand and supply issues. The impact of lockdowns on the supply side and demand side determinants of maternity care was not explored in the included studies, due to data limitations there is minimal discussion of these factors beyond speculations of causal mechanisms. However, we agree with the reviewer that further analysis on policies, and on restrictions to the supply and demand of maternity care services, would be a valuable and interesting further investigation.

The possible impact of lockdowns on the delivery of services and care-seeking has been expanded on in the discussion accordingly:

“Access to high quality maternity care is a key determinant of maternal and neonatal health, which was considerably impacted by lockdown restrictions^{158,174}. Authors hypothesize about the causal mechanisms in the relationship between lockdowns and ABPOs, such as the influence of lockdowns on the delivery of and access to maternity services. However, due to data limitations and lack of sufficient evidence, the impacts of the lockdowns on the supply and demand of maternity care could not be explored in greater detail in this review. Nevertheless, surrounding literature indicates that in HICs maternity services largely remained open throughout lockdowns, but services were altered to protect patients and practitioners from infection risks; care alterations in HICs included remote consultations via telephone and home blood pressure monitoring^{174,175}. During the pandemic there were decreases in antenatal visits and screening uptake, indicating that the lockdowns substantially affected care-seeking behaviour among pregnant women¹⁷⁴. It appears the impact of these services adaptations varied according to an individual's income

level and resource¹⁷⁴. The multifaceted impact of the pandemic and lockdowns on maternity service provision and care-seeking behaviour requires further investigation to ascertain how these care-related factors may have influenced ABPOs.”

Reviewer #2:

Remarks to the Author:

- This was an interesting systematic review that included a large number of papers and concerned an interesting topic, but I wondered at the strategy of focusing on differences in maternal and birth outcomes within high income countries only, where the variation in COVID-19 lockdown impacts would be most difficult to demonstrate, given the existence of well-established and (generally) high-quality national health care systems.

Response to Reviewer #2 remarks:

We thank the reviewer for the positive feedback. The reviewer raises an important issue; however, we and other researchers have found there was high variability between and within HICs in terms of lockdown restrictions. This variation in lockdown restrictions contributed to the authors decision to focus on HICs where, as the reviewer stated, national health systems are well established and generally of high quality. This decision was intentional to reduce potential confounding factors; the similarity of health systems between HICs, meant that the impact of lockdown restrictions on ABPOs and variations in lockdown restrictions by region could be analyzed more clearly.

The variability in lockdowns was considered very carefully by authors, each country's lockdown response was compared and countries which did not have 'stay-at-home', non-essential service closure, or work from home limitations were not included in the study e.g. Sweden. The authors also acknowledge that not every country had a nationalised approach to the pandemic e.g. the USA. To mitigate against this variability, we conducted analysis by continental region as well. There is considerable variability between countries, in lockdown responses, COVID-19 viral infection, morbidity, and mortality rates, and health care systems. In the overall meta-analysis of preterm birth in lockdown compared to pre-lockdown, the I^2 or heterogeneity between studies was high at 96.1%. Authors tried to mitigate this variability as much as possible by restricting to HICs, defining lockdown approaches as comprehensively as possible, and conducting regional sub-group analysis. In doing so, we tried to hold as many potentially confounding factors constant, as much as possible, to explore the impact of lockdowns on ABPOs. Subgroup analysis by continental region reduced I^2 or heterogeneity between studies to 77% in the Europe subgroup, 68.5% in the Middle East subgroup, 55% in the Asia subgroup, and 0% in all other subgroups (North America, Australia, and South America).

Reviewer #2, comment: 1 *The rationale for the study was clear but the discussion was not clearly linked to the rationale. In the introduction, the importance of identifying inequitable outcomes among SE groups with countries was emphasised. The fact that the analysis was not able to identify the expected differences should have been fully discussed and explained in the discussion section, situating this finding within the broader literature on health inequality.*

Specific comments:

Outcomes according to 'ethnicity' and 'deprivation' were the subject of sub-analyses, but these results were not discussed. As this might be considered by many to be an unexpected finding, some explanation is required beyond the discussion of heterogeneity in the limitations section.

Response to Reviewer #2 comment 1 & specific comments:

We addressed this concern by adding relevant text to the discussion section and revised throughout to try and link the discussion clearly to the rationale.

The following text was added to the Discussion:

“Our review provides an assessment of the association of lockdowns with ABPOs and inequalities in HICs. We provided evidence that PTB was associated with decreases during the first lockdown (by 4%) even after accounting for temporal trends (by 3%) which is consistent with other reviews. However, we provide evidence that the associated decrease was restricted to women of white ethnicity, those from least deprived backgrounds, and evident only in Europe and Australia.^{3,6,7} PTB showed a decreasing trend among women living in medium deprivation settings. We showed that lockdowns were associated mainly with decreases in spontaneous PTB and were similar across gestational ages (Supplementary Material: Figure 3: Panel C).

.....

Investigation of inequalities in ABPOs between ethnicity and deprivation groups was limited due to data availability and heterogeneity between studies and settings. Very few studies reported data stratified according to ethnicity and deprivation, and fewer used the same ethnicity groups or deprivation levels. This is understandable as ethnicity distribution varies between settings, and deprivation indicators such as maternal education and neighbourhood income are contextually specific. We addressed this limitation pragmatically by synthesizing available data on deprivation or by using proxy variables for deprivation, and harmonising composite and proxy variables to obtain aggregated estimates of deprivation.

In our analysis, only the white ethnicity group showed evidence of a reduction in PTB associated with the lockdown period. Other, Hispanic, and Asian ethnicity groups indicated a decreasing trend in PTB over the lockdown period, however there was insufficient evidence to support an association. The Black ethnicity group indicated no evidence to support a change in PTB rates associated with the lockdown period. Hence, it appears that the association between lockdown restrictions and PTB

differed according to ethnicity, potentially affecting the magnitude of inequalities among ethnicity groups in HICs.

In terms of deprivation, stratified analysis showed that only the least deprived groups had a decrease in PTB associated with the lockdown period. PTB in medium and high deprivation groups did not materially change from pre-lockdown to lockdown period. Area level estimates of deprivation have an inherent limitation of misclassification of individuals, for instance those who live in areas of high deprivation but are not themselves deprived; this non-differential misclassification blunts the gradient of observed differences in deprivation. When these estimates are combined, this is compounded, underestimating the association which biases towards the null. Our results indicate that lockdowns were associated with unequal impacts on PTB according to socio-economic deprivation and ethnicity, however, the magnitude of the effect may be underestimated and analysis using individual deprivation indices is warranted.”

Reviewer #2, comment: 2 *The methods section was well set out and much of the additional information required was available in supplementary files. I did wonder how case-control studies were handled in the analyses. By definition, case-control study participants are selected on the basis of their outcome status, so combining data from case-control studies and cohort studies to develop a pooled estimate does not seem intuitive.*

Response to Reviewer #2 comment 2:

Studies were included and had data extracted if they presented the raw number of total births/total live births and number of outcomes from two defined periods of pre-lockdown and lockdown. If a study presented raw data from the pre-lockdown, and lockdown period, including the number of observed outcomes, livebirths, and total births from each period, it was included. Raw data on the frequency of outcomes, live births, and total births was extracted from studies, and used to estimate overall risk ratios comparing the advent of outcomes during lockdowns compared to a reference period of before lockdowns. Therefore, data was appropriate to combine regardless of if it was defined as a case control or cohort study. Some studies referred to themselves as case-control studies but considered “cases” as births during lockdowns and “controls” as births in the pre-lockdown period. As such, in the strict, sense they were not case-control studies.

The additional analysis of time-adjusted estimates (estimates adjusted for previous temporal trends in the outcome) were the only analysis to combine effect estimates to obtain a pooled estimate. Studies were assessed based on their study design and chosen methods, rather than what the authors identified as the study type, which was in some cases inaccurate. We included studies which were considered “controlled before and after studies” or “quasi experimental studies” and used the

following Cochrane guidelines to inform study selection

(<https://epoc.cochrane.org/sites/epoc.cochrane.org/files/uploads/EPOC%20Study%20Designs%20About.pdf>).

Reviewer #2 comment: 3 *The characteristics of the studies should be summarised in the Results section. e.g., how many countries were included and the relative proportions of those countries who may have been over-represented, the types of study designs used, etc.*

Response to Reviewer #2 comment 3:

The following information on characteristics of included studies was added to the Results section:

“28 countries were included, countries with the most studies were the USA (46 studies), Italy (13 studies), and Canada (11 studies). Europe and North America were over-represented in the data set (59 studies, 45%, and 54 studies, 41%, respectively). There were fewer studies from Asia (10 studies, 7.6%), the Middle East (11 studies, 8.3%), Oceania (5 studies, 3.8%) and South America (1 study, 0.8%). Most studies were cohort studies (114 studies, 86%), followed by cross sectional studies (15 studies, 11%), and two on prevalence proportion studies and a single case control study. Half of studies were based on data from single sites, such as individual hospitals or clinics (66 studies, 50%). 38 studies used regional or state level data (29%), and 28 used national datasets (21%).”

Reviewer #2 comment: 4 *The tables of included studies (appendix D) should include a column for study design (cohort, case-control, case series, etc.)*

Response to Reviewer #2 comment 4:

A column on study design was added to the table of included studies (available in Appendix D).

Reviewer #2 comment: 5 Minor issues:

Line 96 *“In early 2020, the global spread of SARS-CoV-2 (COVID-19) virus prompted governments...” should be rephrased. COVID-19 is the disease caused by SARS-CoV-2, and does not describe the virus itself. e.g., “In early 2020, the global spread of the SARS-CoV-2 virus (leading to the infection, COVID-19) prompted governments...”*

Line 125-6, *“Lockdowns impacted ABPOs differently in HICs compared to LMICs, and within HICs lockdowns impact varied according to region, time-period, and pre-existing inequalities.”*

This might need to be re-phrased as it currently reads that the authors are claiming there is no health or social gradient in lower income countries.

Line 139 "...with the aim of reducing the spread of COVID-19 infection." COVID-19 is infection with SARS-CoV-2, so the word 'infection' is redundant in this sentence

In several places, the authors refer to data as singular rather than plural (e.g., the text should read 'data were' rather than 'data was')

*Line 242-3 "All analyses were conducted in
243 STATA 17.0." Stata should be written in title case not capitals - i.e., Stata is not an acronym*

Line 393 the word 'Covid-19' should be replaced by 'COVID-19'.

Response to reviewer #2, comment 5 Minor issues:

All minor issues which the reviewer identified have been amended accordingly.

Line 125-6 has been amended to read as follows:

"Lockdowns impacted ABPOs differently in HICs compared to LMICs. While inequalities in ABPOs are present in both LMICs and HICs, within HICs, there is accumulating evidence that lockdowns' impact was inconsistent between regions, time-periods, ethnicity groups and deprivation levels."